

# A survey on exponential random graph models: an application perspective

Saeid Ghafouri and Seyed Hossein Khasteh

School of computer engineering, K. N. Toosi University of Technology, Tehran, Iran

## ABSTRACT

The uncertainty underlying real-world phenomena has attracted attention toward statistical analysis approaches. In this regard, many problems can be modeled as networks. Thus, the statistical analysis of networked problems has received special attention from many researchers in recent years. Exponential Random Graph Models, known as ERGMs, are one of the popular statistical methods for analyzing the graphs of networked data. ERGM is a generative statistical network model whose ultimate goal is to present a subset of networks with particular characteristics as a statistical distribution. In the context of ERGMs, these graph's characteristics are called statistics or configurations. Most of the time they are the number of repeated subgraphs across the graphs. Some examples include the number of triangles or the number of cycle of an arbitrary length. Also, any other census of the graph, as with the edge density, can be considered as one of the graph's statistics. In this review paper, after explaining the building blocks and classic methods of ERGMs, we have reviewed their newly presented approaches and research papers. Further, we have conducted a comprehensive study on the applications of ERGMs in many research areas which to the best of our knowledge has not been done before. This review paper can be used as an introduction for scientists from various disciplines whose aim is to use ERGMs in some networked data in their field of expertise.

## INTRODUCTION

Networks are an essential part of everyday life. From the World Wide Web to biological networks, they all shape the connections of the world. There are many examples of the use of networks in various fields and disciplines. Examples of them include social networks, traffic systems, and disease spread networks. The most canonical way of representing a network is a graph. Indeed, not all of the networks' ties are presented with 100% certainty. For example, in a friendship network, the level of friendship is not the same among all individuals or there is always a chance that two friends stop their friendship in the future Further, in some domains, the current snapshots of the network depend on its timestamp where the network's shape might be different if the snapshot has been taken in another time. For example, in a blockchain network, the structure of the network connections is constantly changing. Hence, the graph has a dynamic structure over time.

Corresponding author
Seyed Hossein Khasteh,
khasteh@kntu.ac.ir

All of these suggest some level of uncertainty in many real-world networks. Therefore, simple graph theory will not suffice for examining these networks. These limitations have led to proposing completely new statistical approaches for graph analysis. More specifically, we want to build a statistical model based on the observed dataset. In these types of graph analysis, a probability in the interval of [0,1] is assigned to each graph. If this probability is close to zero, it indicates that the graph has no chance of existence, while one suggests that this particular graph will undoubtedly exist in the generated data. Any other value between zero and one indicates the existence probability of that graph. These probabilities have different meanings depending on the domain of the network. However, the probability of graph's existence is the most fundamental definition, to which we will stick for the rest of the article.

Statistical graphs (*Frank, 1981*; *Robins et al., 2007a*; *Goldenberg et al., 2010*) have attracted scientists from different disciplines. There are different kinds of approaches regarding their formulation and learning methods. People from mathematics, computer science, physics, and of course statistics have proposed different algorithms and methods for designing the framework for statistically modeled graphs. In addition, statistical graphs are also fundamental to generative models for generating new graphs with similar statistics and attributes to the original graphs. These artificial generated models have various applications, e.g., data augmentation for learning systems where we have datasets with limited resources or simulating and predicting other possible graphs with similar properties. Furthermore, there are longitudinal (*Holland & Leinhardt, 1977*; *Koskinen & Snijders, 2007*; *de la Haye et al., 2017*; *Block et al., 2018*) models which aim to observe a network over a time period and predict the network's future dynamics.

Although different approaches exist, in this work, we are going to review research articles about a particular family of statistical graphs known as Exponential Random Graph Models, abbreviated as ERGM. Designing a statistical model consists of three steps: (1) Designing a general formulation based on the context and statistical specification of the dataset; (2) estimating the parameters of the designed model via some learning methods, where sometimes this step is addressed as the phase of fitting the model to the data; and (3) employing the model with learned parameters to predict the future or unseen part of the data, generation of new data with similar properties, or any other possible tasks. The model utilized for ERGMs (step 1) is almost similar across the entire literature. However, the parameter estimation step (step 2) differs case by case. Figure S1 demonstrates the mentioned steps' flowchart.

The focus of the seminal works such as *Erdös & Rényi (1959)* was mostly on independent tie formation between two nodes. In ERGMs, more complex structures with a reasonable level of dependence have also been taken into account. This approach has led to more complicated models which also require more sophisticated learning methods. Additionally, due to the better accuracy of the models with dependent structures, they are applicable to a more considerable extent of the problems. Therefore, there is a rising interest in using ERGMs in multiple research areas.

Previous surveys (*Anderson, Wasserman & Crouch, 1999*; *Pattison & Wasserman, 1999*; *Robins, Pattison & Wasserman, 1999*; *Goodreau, 2007*; *Robins et al., 2007a*; *Robins et al.,*

*2007b*; *Fienberg, 2010*; *Goldenberg et al., 2010*; *Chatterjee & Diaconis, 2013*; *Chatterjee, 2016*) have introduced most of the articles up to 2016. There have also been two novel surveys in 2018 (*Amati, Lomi & Mira, 2018*; *Van der Pol, 2018*) with a focus on the theory and applications of ERGMs. However, there is a relative paucity of studies investigating ERGMs seminal and new methods together. In addition, to the best of our knowledge, no research has been found examining applications of ERGMs in different fields and contexts in the way that we have done. We believe that this review paper can help the scholars of different disciplines to better recognize the recent applications of ERGMs in their specific field of interest. Certainly, here is still room for more applications of ERGMs in other fields which are yet to be discovered.

There are also some other generative models for network generation such as the use of the neural network for graph generation (*Bojchevski et al., 2018*; *You et al., 2018*) and Stochastic Actor-Oriented Models (*Snijders, 1996*). However, to the best of our knowledge, ERGMs are one of the oldest methods that have been extensively used in the literature up to now.

Several statistical learning methods have been used for ERGMs parameter learning. In this article, we have addressed the following:

- Importance sampling
- Stochastic approximation
- Some of the newly presented methods.

We have introduced some applications of random graphs in the following categories:

- Medical Imaging
- Healthcare applications
- Economics and management
- Political science
- Missing data and link prediction
- Scientific collaboration
- Wireless networks modelling
- Other applications.

Also, some useful tools and libraries have been introduced for the estimation of ERGMs:

- PNet
- R package Statnet
- Bergm.

'Survey Methodology' is a brief description of the methodology we used to find the articles that we believed are related to the topic of this manuscript. In 'Precise Definition of ERGMs' we are going to give a formal definition of ERGMs for the readers who are new to this concept. For experienced researchers in the field, this can be used as a refreshment.

Hence, in 'Methods for Estimation', most of the state-of-the-art works for ERGMs estimation methods have been discussed. 'Preliminaries' is a review of ERGMs' applications

in multiple fields. In 'Applications of ERGMs', we have introduced some of the state-of-the-art new libraries and tools for ERGM estimation. Ultimately, in 'Conclusion', we conclude what we had said and also give some ideas for future works in the world of ERGMs.

## SURVEY METHODOLOGY

For the purpose of finding related research articles we used two different approach.
1. Searching related keyword in the google scholar search engine.
2. Starting from an initial pool of articles and then move back and forth between their citations and references.

In the first approach we search related keywords like ''ERGM'', ''Exponential Random Graphs'', ''Exponential Random Graph Models'' in the google scholar search engine and extracted related articles by reading their abstracts.

In the second approach which was our main methodology throughout the work we initiate with a number of seminal works which were found by one of the following ways.
1. Being introduced by experts in the field.
2. Extracted from the well-known surveys (*Anderson, Wasserman & Crouch, 1999*; *Robins, Pattison & Wasserman, 1999*; *Pattison & Wasserman, 1999*; *Goodreau, 2007*; *Robins et al., 2007a*; *Robins et al., 2007b*; *Fienberg, 2010*; *Goldenberg et al., 2010*; *Chatterjee & Diaconis, 2013*; *Chatterjee, 2016*; *Amati, Lomi & Mira, 2018*; *vander Pol, 2018*) and the well-known book (*Lusher, Koskinen & Robins, 2012*).
3. Papers extracted from the first approach which had a good citation count or were published in journals with high impact.

After finding the initial seed of articles by one of the mentioned methods we checked the related publications that they have referenced and the publications that they have been cited from them. We continued until there were no more related articles. In situations which there were too many related articles our selection criteria were mostly based on the citation count and the journals' impact factor.

## PRECISE DEFINITION OF ERGMS

In this section, we give a brief overview of the overall ERGM scheme. According to *Snijders et al. (2006)* and *Robins et al. (2007a)*, the first work that categorized ERGMs as a separate field of study was (*Frank & Strauss, 1986*). Although it was named as Markov graphs at that time, basically it had the same characteristics. An interested reader can refer to *Robins et al. (2007a)* and *Lusher, Koskinen & Robins (2012)* for more details on both the history and mathematical background of this topic.

In an ERGM, each graph is associated with a probability. This probability indicates the possibility of the presence of that particular graph in the probability distribution of a class of graphs. There are also two other essential elements in ERGMs known as graph configurations and their corresponding parameter. Each configuration or statistics (we will use both names throughout the text) is composed of some nodes and ties repeated in the graph. For example, a triangle consisting of three nodes and edges can be assumed as a configuration. The authors of the seminal work (*Frank & Strauss, 1986*) were the first who

**Table 1  Notations used throughout this work.**

| Notation | Meaning |
|---|---|
| $X$ | The set of all possible graphs with the same number of nodes. |
| $x$ | The variable that indicates the presence of a particular graph from the distribution. |
| $P$ | The probability distribution function of graphs. |
| $S$ | The set of all network statistics presented in the model. |
| $s$ | Some particular statistics of the network. |
| $C$ | The set of all count function of the network configurations. |
| $c$ | The count function of some particular statistics of the network. |
| $\Theta$ | The set of all network statistics coefficients. |
| $\theta$ | Some particular statistics' coefficient of the network. |
| $N$ | The normalizing factor, the sum of all configurations. |

argued that these configurations can be considered as sufficient statistics for a log-linear mode. Sufficient statistics are features of a i.i.d dataset which are sufficient for modeling the distribution probability of the data such that adding another feature does not add any more insight to the model (*RA Fisher, 1922*). So, ERGMs are a representation of the graphs by their configurations. A particular exponential function is defined to represent the relationship between these configurations and the probability distribution of the graphs. This formula is a variation of logistic regression which is extended so that it would handle the dependent variable rather than only being applicable to independent variables which are the case for logistic regression (*Lusher, Koskinen & Robins, 2012*). We will use the notations presented in Table 1 throughout our work.

Note that throughout this work, the representation of the graphs is in the form of the adjacency matrix. For example, in a matrix $x$ if $x_{ij} = 1$ it indicates that there is an edge between $i$ and $j$, while if $x_{ij} = 0$ no edge exists between these two nodes.

Using the introduced notation of Table 1, the ERGM probability function can be expressed as follows:

$$P(X = x|\theta) = \frac{1}{N} exp\{\theta_1 c_1(x) + \theta_2 c_2(x) + \ldots + \theta_p c_p(x)\} \tag{1}$$

$N$ is the normalizing factor which is the sum of the probability of all possible graphs computed by Eq. (1), whose formula is as follows:

$$N = \sum_{x \in X} exp\{\theta_1 c_1(x) + \theta_2 c_2(x) + \ldots + \theta_p c_p(x)\} \tag{2}$$

If we summarize the results, this leads to:

$$P(X = x|\theta) = \frac{1}{N} exp\{\theta^T C(x)\} \tag{3}$$

$$N = \sum_{x \in X} exp\,\theta^T C(x) \tag{4}$$

It can be seen in the Eq. (3) that the network configurations are the building blocks of the ERGM formulation. Choosing the correct configurations with the right relation to the network context is central for to the correct estimation of the graphs' distribution. There are two types of network statistics: (1) statistics based on the edge formations, (2) statistics that are based on the node attributes. In the rest of this section, we are going to introduce some basic network configurations (*Snijders et al., 2006*; *Robins et al., 2007a*) which have been used in the literature.

Structural configuration refers to the statistics that depend solely on the structure of the graph. Note that their usage is not dependent on the network context and can be applied to any networks. These structures are different in undirected and directed networks.

Some structural configurations that are widely used for undirected and directed graphs are presented in (Tables S1 and S2), respectively.

Although use of nodal configurations in our model will cause to be more dependent on some specific context, sometimes it is still useful to leverage this kind of network attributes. The reason is that, in many networks, there is a treasure of useful features in the node's metadata and it is not wise to ignore them as one of our model features.

Some descriptions of nodal configurations that are widely used for undirected and directed graphs are reported in (Tables S4 and S5), respectively.

According to *Morris, Handcock & Hunter (2008)*, there should not be a linear dependence between the configurations that are used in a model. It is due to that fact that the configurations with linear interdependence with each other cannot add any new benefit to the model and only make the model more complicated.

*Snijders et al. (2006)* gave a generalization of ERGMs and also introduced some new configurations. Since then, it has been extensively used in other works. Here we present a brief description of each of them.

Geometrically Weighted Degree Counts (GWDC): this measure is an extension of the nodes' degree combined with geometrically degree discounts in the computation of the statistics, which is expressed as the following expression:

$$GWDC(x) = \sum_{k=0}^{n-1} e^{-\alpha k} d_k(x) \tag{5}$$

In this equation, $x$ is the matrix we want to compute its corresponding GWDC value and $n$ is the number of nodes in the graph. $d_k$ represents the number of nodes with degree $k$. Also, $\alpha$ is a decaying factor which ensures that the nodes with higher degrees have higher impacts.

Geometrically Weighted Stars Counts (GWSC): this measure is an extension of star counts combined with a combination of geometrically degree discounts in computing the statistics, which is expressed as the following expression:

$$GWSC(x) = \sum_{k=2}^{n-1} (-1)^k \frac{S_k}{\lambda^{k-2}} \tag{6}$$

[1] $(d)_r = d(d+1)\ldots(d+r-1).$

In this equation $S_k$ is the number of stars with the $k$ number of edges ($k$-stars). Also, $\lambda$ denotes a decaying factor which ensures that the stars with a higher degree have a greater impact.

Sum of Ascending Factorial Degrees (SAFD): first presented in *Handcock & Jones (2004)*, it is a variation of Yule distribution using the sum of ascending factorials of degree [1] :

$$SAFD(x) = \sum_{i=1}^{n} \frac{1}{(y_{i+} + c)_r} \tag{7}$$

Transitivity by Altering $k$-Triangles (TAT): this measure is an extension of triangle counts combined with geometrically discounts in the computation of the statistics, which is expressed as the following expression:

$$TAT(x) = 3T_1 - \frac{T_2}{\lambda} - \frac{T_3}{\lambda^2} - \ldots + (-1)^{n-3}\frac{T_{n-2}}{\lambda^{n-3}} \tag{8}$$

In this equation, $T_k$ is the number of $k$-triangles. $\lambda$ represents a decaying factor which ensures that the triangles with a higher degree have a more substantial impact. Figure S2 displays a description of $k$-triangles.

Altering Independent Two-Path (AI2P): this measure is an extension of 2-path with a combination of geometrically discounts in the computation of the statistics, which is expressed as the following expression:

$$AI2P(x) = U_1 - \frac{2}{\lambda}U_2 + \sum_{k=3}^{n-2}\left(\frac{-1}{\lambda}\right)^{k-1} U_k \tag{9}$$

In this equation, $U_k$ is the number of star $k$-independent 2-paths. $\lambda$ represents a decaying factor which ensures that the triangles with higher degrees have higher impacts. In Figure S3, you can see a description of $k$-independent 2-paths.

The authors of *Wilson et al. (2017)* addressed one of the significant drawbacks of ERGMs. As can be seen in Tables S2 and S3, the weights of the graphs are missing. In other words, they are only applicable to unweighted graphs, and if we want to use them in the context of the weighted graphs, their weights should be omitted. However, much useful information underlies the weight of the graphs and for most of the domains it is crucial to consider them to accurately model the graph. Following this idea previously discussed in *Desmarais & Cranmer (2012)* and *Krivitsky (2012)*, they continued to design more flexible estimation methods for the so-called Generalized ERGMs (GERGM). Their method can handle a wide range of graphs' statistics with continuous-valued edges.

The endogenous statistics need to be selected before implementing the model. Therefore, there must be several assumptions about choosing a particular statistic. Although the process of finding the best statistics for the model is highly empirical, considerations when making a hypothesis about the network's configurations is important. The choice of a specific statistic is highly dependent on the assumptions we have about network phenomena. Simple structures like the number of edges and nodes take control of the size and sparsity of the graph. In a friendship network, triangles can indicate the inclination of mutual friends becoming friends with each other. In a citation network stars refers to

a large number of central nodes (*Van der Pol, 2018*). The dyadic dependence assumption between nodes should also be considered while choosing the proper statistics for the model. Dyadic dependence is the dependent processes among two dyads. A dyad in this context refers to a pair of nodes and their relation. The dyadic dependence among processes could arise a number of problems like model degeneracy, for more information see *Handcock et al. (2008)*. New specifications like geometrically weighted degree counts and Altering $k$-Triangles have been introduced to alleviate model degeneracies resulted from dyadic dependence. This is achieved by increasing the stability of the model with weighting the low density and reducing the weight for higher degrees to avoid the degeneracy (*Snijders et al., 2006*; *Van der Pol, 2018*).

## METHODS FOR ESTIMATION

One crucial step in the ERGMs models is to fit the coefficient of the model to the observed data after designing the model with desired configurations. Multiple methods exist for this purpose. Nevertheless, the overall approach in all of them is developing a likelihood function based on the ERGM formulation and then solving it with some of the mathematical methods that exist for Maximum Likelihood Estimation (MLE). Note that all of the MLE solution methods should be specialized for the ERGM modeling. After introducing the general form of the mentioned likelihood function, we are going to present a brief description of some of the methods for solving it already presented in the literature.

### A form of the likelihood function

We aim to find the best values of the $\theta$ vector in Eq. (3) which maximize the probability over the observed data. In a more formal expression, we want to solve the following equation:

$$\theta_{ML} := argmax_{\theta \in R^k} P(X|\theta) \tag{10}$$

where, $P$ is the same probability function as the Eq. (3) and, $R^k$ represents all possible real values over a $k$-dimentional space. Note that the $\theta$ is a vector of coefficients rather than a single value; thus, its space value should be a vector space. Different methods exist for solving such equations. Here, we are going to name a few of them which are mostly used in the ERGM related works. Also, we intend to present a number of state-of-the-art methods that have been presented after 2016.

## PRELIMINARIES

### Sampling methods

There are two important applications of sampling methods in the ERGMs parameter estimation model:

- In all methods, there is a need to simulate graphs from the fitted model or simulate some graphs to gain more insight into the distribution of the graphs and their configurations (*Lusher, Koskinen & Robins, 2012*). This distribution is also used to test whether the distribution of the fitted model is close to the observed data or not.
- Predicting the prior distribution of the graphs for Bayesian learning models

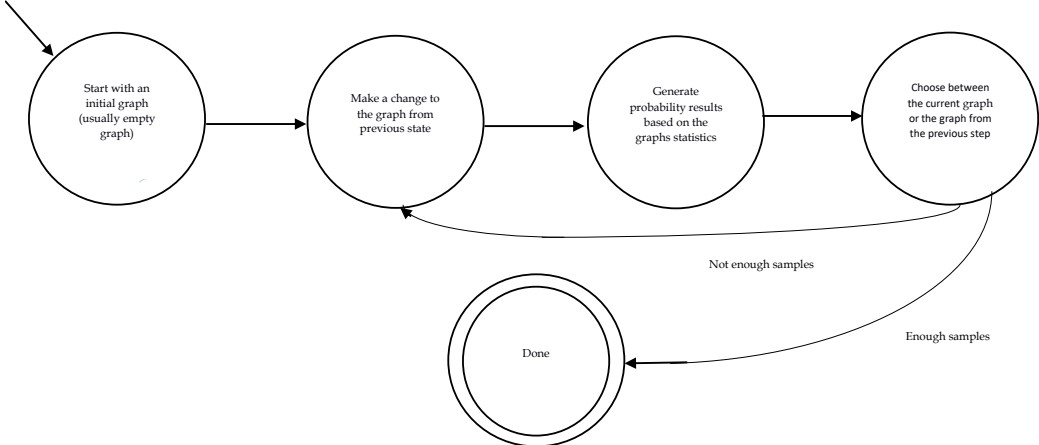

**Figure 1** Finite state automata of a MCMC procedure.

So, there is a need for sampling methods to draw a sample from the given graph distribution. In this section, we present some of the sampling methods that have been used extensively in the literature.

Monte Carlo Markov Chain sampling method which is abbreviated to MCMC (*Metropolis et al., 1953*) is a well-known sampling method which has been used in many works. Here, we only discuss it in the context of graph generation. In this method, we start with an initial graph which can also be an empty graph. Then, in each iteration, a new graph is generated by making a small change to the graph from the last step. The form of this "change" is different from work to work. The most straightforward change is adding or removing a tie. The procedure is as follows: two nodes are chosen randomly. After which the state of their connection is altered (if they are already connected, they become disconnected while if they are not connected, they become connected.). In the next step, the probability of the generated graph is computed according to Eq. (3). This probability is compared to that of the graph generated in the previous step. Then, we accept or reject the new graph based on the comparison of these two probabilities. If the new graph is more probable, it is more likely to substitute the old graph in the next iteration. The probability of whether the new graph is chosen for the next iteration or the graph from the last step is re-chosen depends on which one of them has a higher probability score in Eq. (3). Note that only having a higher probability score is not a guarantee that the graph gets chosen. It only increases the chance of selection. All these outlined the scheme of all MCMC methods. However, the details including how many of ties are altered in each iteration or the probabilistic selection between the old graph and the new one are different in literature. We intend to present a quick introduction to the Metropolis-Hasting sampling methods which is mostly used in ERGM related literature. Figure 1 displays the overall procedure of an MCMC method.

Metropolis-Hasting *Metropolis et al. (1953)* is the most widely used MCMC derivation in ERGM studies. Metropolis Hasting in the context of graph generation is as follows.

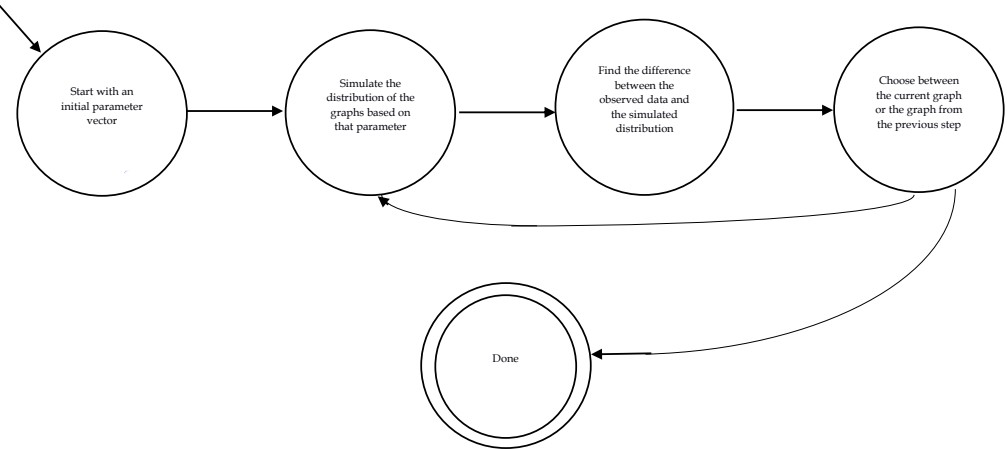

**Figure 2** Procedure of fitting ERGMs variables.

Initially, as we explained in the general MCMC scheme, we start with an empty or random graph. Our goal is to generate $N$ samples from the distribution of graphs, implying that we want to generate a sequence of $x_1, x_2, \ldots, x_N$ graphs. We choose two random nodes at each step and change the tie situation between them. The probability of the newly generated graph and the graph from the last step is then computed using the following formula:

$$\min \left\{ 1, \frac{P(X = x_{new}|\theta)}{P(X = x_{N-1}|\theta)} \right\}. \tag{11}$$

This formula computes the probability of whether to accept the new move or substitute the last step graph as the new one.

## Classic methods

So far, we have reviewed the necessary preliminaries. Now, we can review the most widely used methods in the literature for estimating the value for statistical parameters ($\theta$ in Eq. (4)) best representing the observed data. In other words, our aim is to solve Eq. (10). Most of the methods use the following steps: initially, they start with an initial value for the parameter vector. Then, the distribution of the graphs is generated by one of the sampling methods. Next, the difference between the distribution and the observed data is computed ($E_\theta (C(X)) - C(X_{observed})$). If the difference is satisfactory, the learning process is halted and the current vector of the parameter is considered as the final answer which best fits the observed data; Otherwise, based on the learning method the algorithm moves to the subsequent values of $\theta$ and goes back to step 2. Figure 2 demonstrates the finite automata of this method.

The ultimate goal of all learning methods is to find a vector of $\theta$ values in Eq. (3) that can also generate graphs which are similar to the observed graphs. To this end, different learning methods exist and this section describes the most important of them namely importance sampling and stochastic approximation. We used the description presented in *Lusher, Koskinen & Robins (2012)*.

## Importance sampling

The goal as we said is to minimize the expected value of $\theta$ which minimizes the expected value between observed statistics and the ones generated by the ERGM model. The aim is to use Maximum Likelihood that we discussed before to find the best value of vector $\theta$ which maximize the right hand side of Eq. (10). One possible approach is to search over all possible $\theta$ values in the search space and try them one by one. But since the search space is very large and $\theta$ values are continuous this approach is not practical. Instead of such brute force algorithm, one of the methods for ERGMs parameter estimation is the one inspired by the general framework ML estimation method for dependent data introduced by (*Geyer & Thompson, 1992*). The main idea is to instead of generating the whole possible graphs of a particular $\theta$ vector, we can draw a large sample of the graphs and consider it as a representation of the whole possible graphs at each iteration. This sample is generated from the current value of the $\theta$ vector using the Eq. (3) and is used in the formula to compute $E_\theta(C(X))$ and then compute how much the value $E_\theta(C(X)) - C(X_{observed})$ is close to zero. At each iteration an average over the generated graphs statistics is computed to measure $E_\theta(C(X))$ and decide whether to continue the estimation or not based on how much the $E_\theta(C(X)) - C(X_{observed})$ is close to zero. Other than the mentioned halting situation we need an algorithm to move from each $\theta$ vector to a new one (if the halting is not satisfied). A Newton-Raphson formula is used to move from one statistic to another. For more detail on the mathematical details of the sampling and the used Newton-Raphson based method see (*Lusher, Koskinen & Robins, 2012*).

## Stochastic approximation

The This model presented in *Snijders (2002)* can handle both bimodal and multimodal and enhance the speed of convergence. They also used the Newton-Raphson method for the learning step of the algorithm. As mentioned in *Lusher, Koskinen & Robins (2012)*, they used a three-step method. At phase one, a limited number of iterations is performed to determine initial values of the algorithm. In the second step, the Newton-Raphson algorithm is employed to optimize the answer. Finally, the convergence criteria are checked.

## Newly presented methods for ERGM estimation

In *Byshkin et al. (2016)*, the authors improved the MCMC sampling part of the ERGM estimation by adding an auxiliary parameter to the model. In their method, which they called Improved Fixed Density (IFD) MCMC sampler, they tried to decrease the state space of the network to reduce the time complexity of the algorithm. This new auxiliary variable which was based on the number of ties helped the model to converge faster without the need of making the MCMC overall model more complicated.

In some works like (*Stivala et al., 2016*), snowball sampling (*Coleman, 1958*; *Goodman, 1961*) was used to overcome the computational complexity of the MCMC method over large network datasets.

As mentioned earlier, the Bayesian estimation of the parameters requires prior knowledge about the network posterior distribution. However, this posterior probability distribution

is not always easily available. To overcome this issue, (*Bouranis, Friel & Maire, 2017*) introduced a pseudo-likelihood estimation approach by replacing the posterior distribution with a more achievable pseudo-distribution. Although this method resulted in faster computation of the likelihood function, as mentioned in the (*Schmid & Desmarais, 2017*), its results are not still as precise as they should. To handle this problem, the same mentioned article introduced another pseudo-likelihood estimator based on the bootstrapping parameters which culminated in more accurate convergence.

In a recent work (*Bouranis, Friel & Maire, 2018*), the authors proposed yet another heuristic model based on pseudo-likelihood estimation. They did so by performing three adjustments to the pseudo-likelihood function: (1) mode corrections to overcome the bias of the pseudo-likelihood function; (2) curvature adjustment, which is a modification in the selection of the transformation matrix and the corresponding Hessian matrix; and (3) magnitude adjustment, which is a linear transformation to scale the curvature-adjusted pseudo likelihood to the right values.

Despite all the progress in the ERGMs parameter estimation and modeling, it is still a hard task in large graphs. *Thiemichen & Kauermann (2017)* addressed two of the main challenges of ERGMs, including the instability of the model especially in the models with more straightforward statistics like triangles and the time-consuming nature of the ERGM parameter estimation procedure due to large number of numerical simulations. For solving the first problem, they proposed a technique to produce smooth stable statistics. Further, to overcome the second issue, they employed a novel subsampling model which instead of fitting the model to the whole network it only fit the model to subgraphs from the network and then aggregated these sample estimates. The two mentioned ideas yielded a significant improvement for modeling large graphs.

## ERGMs variations

Apart from the basic definition of ERGMs there are also some other variations of ERGMs. Each year a number of new extensions of the original ERGM definition are introduced. In this chapter we introduce three of the most widely used ERGMs variations.

Evolution of networks in dynamic environment like social networks has attracted scientist to make an extension of the ERGMs called Temporal ERGMs a.k.a. TERGMs which is capable of capturing the information underlying dynamics of such networks (*Hanneke et al., 2010*). A Markov assumption between snapshots of the network at each timestep is taken. Then the model is created based upon the relation between each two consecutive snapshots $S_t$ and $S_{t-1}$.

$$P(X = S_t|S_{t-1}, \theta) = \frac{1}{N(\theta, S_{t-1})} exp\{\theta^T, \psi(S_t, S_{t-1})\}. \tag{12}$$

As it can be seen in Eq. (12) most parts of the formula for TERGM is similar to normal ERGM. However, the time snapshots are now considered and each new time snapshot $S_t$ is dependent to its previous one $S_{t-1}$. Also, the normal count of the networks statistics has been substituted with temporal potential count $\psi$ over two consecutive snapshots. For more information see *Hanneke et al. (2010)* and for the information about the btergm which is a library for temporal ERGMs see *Leifeld, Cranmer & Desmarais (2017)*.

Most of the real-world network are associated with a value on their edges which are referred to as weighted graphs in graph theory. A plethora of researches have been done to consider these types of networks into the ERGM general schema. GERGM (*Desmarais & Cranmer, 2012*) and the model proposed by *Krivitsky (2012)* are the two most well-known models which have incorporated the networks' edges' weights into the model. The normalizing factor in Eq. (3) which is the denominator of Eq. (4) is not assured to be convergent when the network statistics ($C(x)$ in the Eq. (4)) are infinite set like continuous valued edges. GERGM is a model aimed to overcome this issue by using a probability model for such continuous values. They build a transformed version of the original ERGM formula that no longer suffers from the mentioned problem. The *Krivitsky (2012)* have also extended the previous binary version of ERGM which only models edges existence rather than their value into a model which is capable of capturing the information of weighted graphs. However, his method is restricted to natural valued weights on the edges.

In network science there is a special kind of networks called multiplex or multilayer networks. These are networks which their nodes are connected in the context of more than one attribute. For example, in a social relation network, actors might have several relations between them like friendship network or co-working network. Each of these relations can be abstracted as a layer in a network model. Also, in some situations, there is a hierarchical structure in the data like modeling the relations inside a university. There are schools, which are divided into groups and lecturers and students. An extension of ERGM which is applicable to model such scenarios in multilevel networks is proposed for these networks (*Wang et al., 2013*). They considered relation between the nodes in each level and also the inter-level relations into the model. For example, consider a two layer network with layers $A$, $B$ and an imaginary layer between them called $x$ which is for the purpose of modelling inter-level relations between $A$ and $B$. Then the Eq. (1) is re-written as:

$$P(A = a, X = x, B = b|\theta) = \frac{1}{N} exp\{\theta_a^T C_a(a) + \theta_b^T C_b(b) + \theta_x^T C_x(x) +$$

$$\theta_{a,x}^T C_{a,x}(a,x) + \theta_{b,x}^T C_{b,x}(b,x) + \theta_{a,b,x}^T C_{a,b,x}(a,b,x)\} \tag{13}$$

Which the $\theta_a^T, \theta_b^T, \theta_x^T, \theta_{b,x}^T, \theta_{a,x}^T, \theta_{a,b,x}^T$ are the parameters for statistics which are extracted from layers $a, b$ and the inter-level relations $a, x$ and $b, x$ and the inter-level relation of layers $a, b$. The same is true for the count functions of the statistics. $\theta$ is the set of all types of statistics.

## APPLICATIONS OF ERGMS

As mentioned previously, ERGMs are a useful tool for scientists from various disciplines. Networks are everywhere, and anywhere that they exist they can be analyzed using ERGMs and other statistical models. Note that here we have mostly reviewed the works since 2016.

### Medical imaging
In order to take care of the limitations of the descriptive analysis of brain neural networks, the author of *Sinke et al. (2016)* used ERGMs to be able to model the observed network using the joint contribution of network structure. They also compared the changes in brain

networks statistics across different ages. This study was conducted to examine the effects of aging during lifetime in the brain global and local structures. Graphs where extracted from brain images obtained from diffusion tensor imaging (DTI). Four network statistics were used to model these networks:

- The number of edges
- The geometrically weighted edgewise shared partner (*Hunter, 2007*)
- The geometrically weighted non-edgewise shared partner (*Hunter, 2007*)
- The hemispheric node match: a binary indicator which shows whether two nodes are in the same hemisphere of the brain.

The Bayesian learning schema from *Caimo & Friel (2011)* was used to fit the model.

In a recent work, (*Dellitalia et al., 2018*) employed ERGMs to study the structure of neural networks of the brain. They aimed to increase the chance of unconscious and injured patients to recover by analyzing brain functional data. In their work, they overcame four shortcomings of previous methods by incorporating ERGMs into their study. For example, one of them was the ability to assess the dynamics of the network over time.They used the Separable Temporal ERGMs (TERGM) (*Krivitsky & Handcock, 2014*) for their modeling. One of the aspects of their work that successfully handled with ERGMs was that the network structures they chose should have not been necessarily independent. This restriction was one of the main drawbacks of previous methods.

Functional Magnetic Resonance Imaging or fMRI is a method for observing brain activities and their changes over time. There are components in the fMRI images which can be explained using network analysis methods. Nodal signals, network architecture, and network function are the three essential network properties in building fMRI-based networks (*Solo et al., 2018*). ERGMs are one the main important network analysis methods which have been used to explain such networks. The authors of a review paper (*Solo et al., 2018*) introduced the most critical efforts with the aim to explain these brain networks. Note that there are plenty of works which used ERGM as their method (*Simpson, Hayasaka & Laurienti, 2011*; *Simpson, Moussa & Laurienti, 2012*).

## Healthcare applications

Having a healthy life is one the central concerns of human life. If we look at this issue from a macro perspective, we can see that many health-related problems can be alleviated by analyzing their corresponding inter-related actors. For example, in epidemiology, there is a direct connection between the patient relationships and the extent that the disease can spread. In most cases, these relations between the actors will result in the formation of a network. This network can be analyzed using ERGMs to answer different questions underpinning its formation and dynamics. This kind of analysis is something that has already been done extensively by researchers in the healthcare community.

Analyzing inter-hospital patient referral network is a significant problem which (*Caimo, Pallotti & Lomi, 2017*) has recently investigated using ERMGs. They used a combination of the edges and nodes of the network and utilized the Bayesian approach introduced in

*Caimo & Friel (2011)* to fit their model. This task was done using BERGM (*Caimo & Friel, 2014*) R language package for their implementation.

Another work (*Baggio, Luisier & Vladescu, 2017*) shed light on the relationship between social isolation and mental health. The connection between these two subjects was investigated by analyzing the network of Romanian adolescents using ERGM modeling. They concluded that there is a strong link between the two mentioned concepts.

Application of statistical network in epidemiology and disease spreading is another interesting topic which has attracted from the attention of the biological science community. (*Silk et al., 2017*) provided an important opportunity to advance the understanding of the pattern and evolution of infections in static and dynamic environments. They also used ERGMs for their models. In their ERGM model, they employed a fair number of both structural and node-based attributes. The ERGM (*Hunter et al., 2008*) R language package was used for the tests.

Social ties can reveal a wide range of aspects of human life. The networks formed by such ties and edges among individuals can transfer life habits and behaviors in a society. For example, in many kinds of literature, the relationship between social tie and analysis of obesity has been investigated. *Zhang et al. (2018b)* thoroughly studied the articles related to applications of social network analysis to obesity. In another work related to eating disorders, (*Becker et al., 2018*) have presented some findings using ERGM network analysis about the relationship between the eating disorders and human relationships. They conducted their study on members of a sorority at Southeastern University.

## Economics and management

Marketing organizations that are responsible for promoting tourist destination have also been analyzed using ERGMs. (*Williams & Hristov, 2018*) intended to study the networks underpinning Destination Marketing Organizations (DMOs). They developed four models with the most complex one consisting of the following statistics:

- Number of edges
- The geometrically weighted edgewise shared partner (*Hunter, 2007*)
- Properties of membership and industry background.

Global migration and different attributes of immigrants can be considered as a network. There are many theories on how these networks shape and evolve and how they depend on immigrants and country backgrounds. ethnicity, wealth, religion). (*Windzio, 2018*) applied ERGM in order to examine theories and hypotheses about creation and evolution of these networks. He used both the graph structure and node attributes in a large number of statistics.

Global tourism and its corresponding network, Global Tourism Network (GTN), is yet another field of study, given the tremendous financial importance of tourism market. As mentioned in *Lozano & Gutiérrez (2018)*, it is essential to gain insight into the connections between its components. In the same article, an ERGM approach was used to find the critical local substructures of the GTNs.

Handling the budget and resources during crises is always a challenging task for humanitarian organizations. There is a need for a tradeoff between the use of asset supplies for the current crises and the usual ongoing projects. This problem has been formulated in the form of asset supply networks. *Stauffer et al. (2018)* used ERGMs as an empirical model to understand the asset flows during a crisis.

The applications of ERGMs have even been extended to the analysis of online drug distribution networks. In a recent work, (*Duxbury & Haynie, 2018*) conducted the mentioned research on a dataset of an online drugstore on the dark web. They studied such networks concerning their topological dynamics, suppliers, and customer demand as well as the resistance of such networks to disruptions.

Does economic partnership between professionals will result in further trust and solidarity? This is the central question of *Bianchi, Casnici & Squazzoni (2018)*. They developed an ERGM multiplex network model collaboration network and a number of other attributes and then analyzed it using multivariate ERGMs to examine social support and trust for each of the network statistics.

## Political science

A large number of articles in the political science community have used ERGMs for their modeling. This enthusiasm toward ERGMs among political science scholars well suggests that it is among the most famous mathematical modeling in the field. Here we introduce a handful of these articles.

Sustainable development policy is a major concern both for the government and the private sector. It is only achievable by interaction among individuals. In particular, the role of the connection between funding sectors and those in need of money is important for carrying out their projects. This is the central problem of *Gallemore & Jespersen (2016)*. The dataset consisted of 91 donor organizations. The role of ERGM in this work was modeling the donor agent relationship networks.

Another major issue that has been addressed through ERGMs is collaborative governance between different sectors and individuals of multiple organizations. In *Ulibarri & Scott (2016)*, the authors used ERGM to test their hypothesis about what should be observed in low-collaboration vs. high-collaboration networks. Four simple ERGMs' configurations were used, including:

- The number of networks ties
- The number of nonzero ties
- The number of reciprocity relations in the network
- The number of transitivity relations in the network.

In a more recent work, (*Scott & Thomas, 2017*) addressed the same problem. However, they used different datasets and network statistics. *Hamilton & Lubell (2018)* also took the same ERGM modeling approach in discussing the collaborative governance, in the special domain of climate change adoption.

In an exciting work, *Li et al. (2017)* investigated the effectiveness of military alliances in making peace between states. They used temporal random graph models for longitudinal

network data of alliance. They employed two different sets of network statistics and developed two models upon them.

Communications via internet social networks have helped the human to take a huge step further. People from multiple backgrounds and societies are engaged in conversations that have never been possible before the widespread popularity of online social networks. In the case of political conversations in social networks, there is always the dilemma whether this freedom has resulted in more communications between people with different ideologies or adversely it will cause people with same viewpoints tend to dominate most of the conversation thereby self-reinforcing the same way of thinking. (*Song, Cho & Benefield, 2018*) addressed this issue by studying the network of message selection of users during a presidential election and then analyzed the mentioned network by a Temporal ERGM (TERMG) to answer the questions above. The world trade network has also been investigated via TERGMs. *Pan (2018)* studied these networks to answer the underlying questions about them and their effects on related subjects.

Even further, some works such as *Chen (2019)* took the use of ERGM networks in modeling political networks a step further by incorporating multilayer networks properties into their models. He proved with experimental results that this multilayer approach toward ERGMs could better fit the model to the observed data.

The analysis and challenges of power transition in a personalized authoritarian system is a problem that has been discussed in *Osei (2018)* using ERGM modeling. In addition to qualitative methods, the author employed ERGM as a quantitative method to answer questions about the regime survival of the regime under the mentioned situations. The network in this context consisted of elite interactions network in authoritarian countries. They found that many of the important people in the past ruler administration still play a crucial role in the current government.

Environmental treaties among governments play a vital rule in solving environmental issues. However, coming to an agreement in such commitments is not straightforward. The aim of *Campbell et al. (2019)* is to study the model of ratification in such treaties among different parties or states. The main contribution of this research is to find out how the influence network between countries can affect the interdependency of countries decisions on environmental politics. To this end, they have used Bipartite Longitudinal Influence Network (BLIN) model to extract two latent influence network using which show negative and positive influence among different countries. Later these two networks have been analyzed using ERGMs to find the effective contextual and structural network statistics on the shaping of influence (negative or positive) networks (*Marrsetal, 2018*).

Network of international arm trade is yet another subject that has been studied using ERGM simulations (*Thurner et al., 2019*). The structure of weapon exchanges network between countries and alias is very complicated. A plethora of effective factors are effective in the formation of the network. Economic enhancement of the seller and the desire to strengthen they allay in different regions of the words are two important considerations from the dealers. Their datasets are extracted from available data of arm deals after world war II. Temporal ERGMs were used for during the analysis. They have used a number of statistics based on their hypothesis about importer and exporter effects, size of the

countries' domestic economic markets, national material capabilities, conflict involvement joint membership in defense agreement, geographic distance between two countries. Most of the statistics used in this work were exogenous statistics.

## Missing data and link prediction

Link prediction is the problem of finding missing links in a network. As we explained, ERGM deals with estimating graph distribution and generating a new graph based on them. Graph generation part is the exact process of finding missing links. However, in link prediction, we do not want to estimate the whole graph distribution, and we desire to find the probability of link formation between two nodes based on the current structure of the graph.

*Smith (2012)* used ERGMs to create a global view of networks with missing data based on sampled data. In their approach, they took sampled ego networks and tried to estimate features of the whole network. The interesting fact about this work was that not both the structure of the network and its size were unknown. A three-step algorithm was used, and in the last step, the aim was to predict the global structure of the network from the fitted model. Two real-world network data were used in the tests including addition of health network and sociology co-authorship network.

*Koskinen et al. (2013)* used the same approach of leveraging ERGM for data augmentation in graphs with missing tie variable. In an empirical test, they were able to estimate the missing tie variable of a network with 74% missing tie with fair precision. As the article name suggests, they used a Bayesian estimation method for fitting the parameters of their model.

*Zhang, Zhai & Wu (2013)* applied ERGMs for predicting links in microblogs. They used five kinds of graph statistics with four of them (2–5) introduced in *Hunter (2007)*:

- Number of edges
- Gwidegree (Geometrically weighted indegree): the weighting indegree of the network.
- Gwidegree (Geometrically weighted outdegree): the weighting outdegree of the network.
- Gwodegree (Geometrically weighted dyadwise shared partner): the number of shared nodes of all node pairs in the network.
- Gwesp (Geometrically weighted edgewise shared partner): similar with Gwdsp, it is the number of shared nodes for linked node pairs in the network.

The link prediction based on the ERGM method introduced in this article is an iterative approach. At each step, they compute the conditional probability of adding an edge between two arbitrary nodes having the observed part of the network. This process is performed several times through an MCMC simulation, and at last the average of all these steps is computed:

$$P\left(X_{ij} = 1 | X^c = x^c\right) = \frac{1}{N} exp\left(\theta^T C\left(x_{ij} = 1, x^c\right)\right) \tag{14}$$

In Eq. (14), $X_{ij}$ is the probability of presence of an edge between nodes $i$ and $j$. $X^c$ is also the state of all other edges in the time predicting $X_{ij}$.

Five datasets have been used:

- Sina Microblog dataset community of "Beijing badminton community."
- Sina Microblog dataset community of "Beijing bicycle community."
- Sina Microblog dataset community of "Data mining community."
- Scientist co-authorship dataset GR (General Category).

The authors of *Krause & Caimo (2019)* have presented a new estimation algorithm for Bayesian Exponential Random Multi-graphs model which is an imputation model applicable to such multi-layer networks. This work is an extension of the *Koskinen, Robins & Pattison (2010)* to multi-layer networks.

An interested reader can refer to a recent survey on different imputation method on network missing data (*Krause et al., 2018*). One of the advantages of this methods is that it is solely about missing data in the context of networks. Different missing data treatment methods have been tested on different missing data in a complete benchmarking framework.

## Scientific collaboration

Finding the best colleagues or best-related research papers and topics is always a significant issue for anyone in the scientific community. Co-author and citation networks analyses are two important topics that have been extensively studied in research related to analysis of networks addressing these issues.

The researchers in *Zhang et al. (2018a)* addressed the effect of three major network properties in scientific collaboration networks including Homophily, Transitivity, and Preferential attachment. Performing an ERGM study on these networks, they argued that incorporating the mentioned properties we can provide more insight into how collaborations form. The data for this study were collected using the metadata of papers' citations from the Web of Science from 1956 to 2014.

As we approach more complex scientific phenomena, we more feel the need for collaboration between different scientific communities. *Fagan et al. (2018)* also studied a co-authorship network to evaluate the changes in inter-disciplinary scientific articles. More precisely, they applied a special form of ERGMs called the Separable Temporal ERGMs (STERGM) *Krivitsky & Handcock (2014)* to evaluate the co-authorship network over time and make prediction ties in the network. They employed some structural and nodal attributes. Structural attributes refer to a number of edges, degree, and triadic closure, while some nodal attributes capture whether two individuals have the same professor rank, gender, and college.

ERGMs are widely used for citation networks analysis. *An & Ding (2018)* performed the same study in the special case of publications on causal inference. They argued that some technical and social processes are underpinning citation networks. Their ultimate goal was to explain the essential factors in forming a citation network and predicting the citation patterns.

An in-depth study of polarization among researchers of the field of social science was performed in a recent work (*Leifeld, 2018*). He used both qualitative and quantitative methods to address the most compelling reasons and strategies causing the polarization.

He applied ERGM as his qualitative method over two co-authorship networks in the field of social science in two separate countries.

Other than the studies on co-authorships and citation networks there are other aspects of scientific collaboration that have been widely studied. One of such studies is the study on how the recruitment of new members of scientific collaborators in scientific organizations takes place. In a study (*Leifeld & Fisher, 2017*) the dynamics underlying the membership procedure of new scientist in international scientific assessments has been evaluated. The authors have used a dataset extracted from an international well-known research program on world's ecosystem called Millennium Ecosystem Assessment (MA). Their method is based on analyzing the pattern of the network formation by ERGM using a number of exogenous and endogenous network's statistics. The analysis approved the authors hypothesis which suggests that factors like having the same nationality to the previous researcher in the research group or being in the same institution with them have a high impact on the recruitment of new researchers. This could result in lack of diversity of opinions in the final outcomes of the assessments conducted by the research group.

## Wireless networks modelling

Random Geometric Graphs also known as RGGs are defined as the group of graphs which are obtained by placing a number of nodes randomly in a geometrical space and draw vertices between those nodes which their distance is less than a threshold d in a given norm (*Penrose et al., 2003*). One issue in the wireless sensor networks is that there is not a fixed placement for the nodes in most of times. The nodes are randomly distributed and therefore the shape of connecting graph tend to be very volatile (*Raghavendra, Sivalingam & Znati, 2006*). Studying these graphs formation and the statistical dynamic behind their formation has been extensively investigated in the literature related to RGGs (*Iyer & Manjunath, 2006*). For example, exponential RGGs which are the RGGs that the distribution of their nodes is also exponential (*Gupta, Iyer & Manjunath, 2008*). These graphs have been used for modeling wireless sensor networks *Shang (2009)* abd *Kenniche & Ravelomananana (2010)*. In the *Shang (2009)* the wireless sensors are assumed to be on a line and to evolve over time with respect to a dynamic RGG process. The effect of statistical properties for a particular time snapshot has also been considered in this paper. Such analysis with using one-dimensional RGG has also been done in the past in *Karamchandani et al. (2006)*. Vehicular Ad Hoc Network are yet another use case for RGGs (*Zhang et al., 2014*). Due to the movement of the vehicles and the rapid changes in the graph they have many similarities to previous applications of RGGs.

## Other applications

The applications of ERGMs are so extensive that some works cannot be organized in a particular category. In this section, we introduce some of them.

The concept of social networks is not limited only to human relationships. There are some complex interactions in animal behaviors which can be modeled as graphs. ERGMs are also a useful tool for analyzing these kinds of networks. In a recent work, (*Silk & Fisher, 2017*) reviewed the use of ERGMs in such studies. Also, more specifically, other recent

works have leveraged ERGMs capabilities in their specific context. *Hellmann & Hamilton (2018)* is a work in which the authors investigated the effect of neighbors' mediation in cooperative fish breeding by analyzing their interactions with an ERGM model. In another work (*Silk et al., 2018*), the same approach was used to investigate sex-related disease spreading through animal contact networks in three sorts of animal networks.

In a novel work, *Müller, Grund & Koskinen (2018)* studied the social inequalities in Sweden by analyzing an immigrant movement flow network on both the micro and macro levels. Their network was a directed binary graph with Stockholm's neighborhoods as the nodes and ties as the representative of the movement flow across neighborhoods. Only structural features (statistics) were used in ERGMs.

How do networks respond to a sudden change? Which sort of disruptions is most influential in the network upcoming status? How the network will react to a change or what is the best reaction? These are all questions that can be summarized as "forecasting social network reaction to disruption." In a recent article, *Mellon & Evans (2018)* reviewed state-of-the-art research articles concerning these topics in various fields. According to them and by mentioning one of their previous works (*Mellon, Yoder & Evans, 2016*), ERGMs can play a crucial role in this issue. In the mentioned work, they used ERGMs to examine the network formation mechanism before and after the intervention. According to their findings, networks tend to preserve these mechanisms following the disruptions.

## TOOLS AND LIBRARIES

There are a number of useful tools and libraries that facilitate use of ERGMs in different domains. PNet and its extension for multilevel networks (MPnet) and bivariate analysis (XPnet) were introduced by *Wang, Robins & Pattison (2006)*. It is a stand-alone software, it has both windows .NET and Java versions. Because of the Java version, it can be considered as a cross-platform application. Also, since it is not a library of some other languages and thanks to its user-friendly environment, it is the most suitable choice for people with less computer programming background. It is also a free software application and can easily be downloaded through its website (http://www.melnet.org.au/pnet/).

Statnet (*Handcock et al., 2003*; *Handcock et al., 2008*) is an R language package which can implement most state-of-the-art ERGM methods and algorithms. It also has a variety of capabilities via other R libraries. For example, some visualization options are available through libraries such as dynamic network and rSoNIA. A wide range of network configurations has been implemented in this package. It has an active community, and it seems that it is the de facto standard library for ERGMs. Thanks to its open source and well-documented codebase, it can be used as a template for implementation of new methods. However, because it is a programming language library and not standalone software, it requires minimum knowledge of programming. *Goodreau et al. (2008)* have presented a detailed explanation for its installation and usage. There are also other extensions for the Statnet; for example, *Caimo & Friel (2012)* has incorporated the Bayesian ERGMs into the library.

# CONCLUSIONS

This study offered an explanation of Exponential Random Graph Models aka ERGMs. We also reviewed some state-of-the-art methods published after 2016. These articles either presented new methods for fitting the ERGMs parameters or studied the possibility of using new network configurations. Further, we did a comprehensive study of the research articles published by scientists of multiple disciplines which have leveraged the applications of ERGMs in their fields of interest. Multiple variation of the ERGM networks have been reviewed. We classified research articles in seven plus one (other applications) categories. These included research works in medical imaging, healthcare applications, economics and management, political science, missing data and link prediction, scientific collaboration, Wireless Networks Modelling, and other applications. Altogether, these studies provided valuable insight into the potential use of ERGMs in interdisciplinary research. We also presented a brief description of the ERGMs tools and libraries which can be used by scientists to conduct research like the research papers we presented. The objective of this study was to develop an understanding of the ERGMs methods and applications for those with limited knowledge about them. However, more in depth study for applications of ERGMs in each special area of study is still needed. These domain specific studies can do further analysis on the technical side of the ERGM modelling which was not a concern of our work. Some potential future directions for future research are:

- There are many good papers investigated the applications of Exponential Random Graphs from social science research community. However, there is a lack of interest among engineering community in these methods. Investigating the possibilities of using ERGMs in networked data in various field of engineering studies is a research path should be considered in the future. Some examples are studies on computer network topology, internet measurement which this statistical tool might be used for prediction of missing links or for the purpose of data, etc.
- Multilayer networks are now widely studied in different disciplines e.g., transport and economical networks. Despite some good works using state of the art ERGMs methods for multi-layer networks there is still a lack of interest in using statistical tools like ERGMs for them comparing to other methods.
- The hype of deep learning (*LeCun, Bengio & Hinton, 2015*) has made many new possibilists for combining them with traditional methods to achieve better estimation. To the best of our knowledge no work has been done to this date trying to leverage graph based deep learning methods alongside ERGMs.
- Despite the existence of comprehensive libraries for ERGMs like statnet, there is still no library for it written in Python. Since Python is the most used programming language in data science it is worthwhile to implement a powerful library for ERGMs modelling in Python. One possible way is to extend current widely used libraries like NetworkX (*Hagberg, Swart & Chult, 2008*) to include ERGMs in them.
- To the best of our knowledge there is no comprehensive research on comparison of ERGMs with newly presented generative graph models like NetGAN (*Bojchevski et al., 2018*).

### Funding
The authors received no funding for this work.

### Competing Interests
The authors declare there are no competing interests.

### Author Contributions
- Saeid Ghafouri conceived and designed the experiments, performed the experiments, analyzed the data, prepared figures and/or tables, and approved the final draft.
- Seyed Hossein Khasteh analyzed the data, authored or reviewed drafts of the paper, and approved the final draft.

### Data Availability
Our paper is a review paper and there is no code or raw data associated with it.

### Supplemental Information
Supplemental information for this article can be found online at http://dx.doi.org/10.7717/peerj-cs.269#supplemental-information.

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
