# Peer review of "A survey on exponential random graph models: an application perspective"

_PeerJ Computer Science, doi:10.7717/peerj-cs.269_

## Round 0.1 · original submission · Major Revisions

The reviewers have raised some concerns on the exponential random graphs. Please address the comments point by point. Exponential random geometric graph should also be compared: Exponential random geometric graph process models for mobile wireless networks.

Reviewer 1 ·

Basic reporting

Is the review of broad and cross-disciplinary interest and within the scope of the journal?

Yes.

Has the field been reviewed recently? If so, is there a good reason for this review (different point of view, accessible to a different audience, etc.)?

Yes, but focuses on the technical perspective, while this emphasizes more on applications.

Does the Introduction adequately introduce the subject and make it clear who the audience is/what the motivation is?

Yes.

Experimental design

No comment.

Validity of the findings

No comment.

Additional comments

This is a review article on exponential random graph models and their applications. I have a few points for the authors to consider.

As the endogenous factors need to be selected before implementing the model, it might be also helpful to introduce some basic considerations when making hypothesis about the theory. For instance, why would a user of ERGM incorporate covariates, e.g., k-star, edge-wise shared partners, and four-cycles, into their model? What statistical effect, significantly postive or negative, will that covariate have on the outcome network? The authors should emphasize the importance of this precedure when choosing endogeneous network effects because they are inherently different from the control variables used for classic statistical models such as logistic regressions.

Authors should also have a more elaborate discussion on recent advances of ERGM, such as temporal ERGM, weighted ERGM, et al., and their potential applications in science.

A couple of other papers on the application of ERGM:

Ecosystem assessment: Leifeld, Philip, and Dana R. Fisher. "Membership nominations in international scientific assessments." Nature Climate Change 7.10 (2017): 730.

Environment: Campbell, Benjamin W., et al. "Latent influence networks in global environmental politics." PloS One 14.3 (2019): e0213284.

Arms trade: Thurner, P. W., Schmid, C. S., Cranmer, S. J., & Kauermann, G. (2018). Network Interdependencies and the Evolution of the International Arms Trade. Journal of Conflict Resolution, 0022002718801965.

Reviewer 2 ·

Basic reporting

The survey is not written in professional English so it has to be re written in most parts. References are covering most of the literature but both theoretical and application context can be explained better.

Instead of citing:
- Caimo A, Friel N. 2012. Bergm: Bayesian exponential random graphs in R. arXiv preprint arXiv:1201.2770.

The authors should cite:
- Caimo, A. and Friel, N. (2014), “Bergm: Bayesian Exponential Random Graphs in R,” Journal of Statistical Software, 61(2), 1 – 25.

Regarding missing data. There are a few more references that can be included:

- Krause, R. W., Huisman, M., Steglich, C., & Sniiders, T. A. (2018, August). Missing network data a comparison of different imputation methods. In 2018 IEEE/ACM International Conference on Advances in Social Networks Analysis and Mining (ASONAM) (pp. 159-163). IEEE.

- Krause, R. W. and Caimo, A. (2019), “Missing Data Augmentation for Bayesian Exponential Random Multi-Graph Models.” International Workshop on Complex Networks, 221, 63 – 72. Springer.

Experimental design

The Section "Computation of prior probability" is not clear and correct. I guess that section should be focusing on Bayesian methods where the target is the posterior distribution.

The section Classic methods also is not clear. This should be Maximum likelihood estimation and should come before the Bayesian section. The estimation methods should be explained in more details especially regarding the MCMC-MLE and network sampling algorithms.

Validity of the findings

The survey is timely and potentially very useful but can be improved in terms of explanation of the results obtained in the literature. Model assessment methods are not present, they should be included. Conclusions should include more discussion of the future directions / challenges.

Additional comments

The survey must be improved a lot in order to be publishable. The text is often not clear or very difficult to understand for those who are not expert in this research field. Much more explanations and details are needed about the practical implementation of the ERG models.

Reviewer 3 ·

Basic reporting

It is suggested to add some tables, to compare each method in detail and make the article more clearer.

Experimental design

It is suggested to cite some studies about engineering. There are some outstanding research by using ERGM on communication, electrical engineering, and so on, done by engineers.

Validity of the findings

There are no clear conclusion presented by authors. It suggested to add some conclusion about what are the appropriate methods for different problems.

Additional comments

The article is clear in logic and accurate in language. It is suggested that engineering-related reviews and citations be added.

---

## Round 0.2 · accepted · Accept

All previous comments have been addressed. It can be accepted.